# Experimental Investigation of Mode Localization’s Bifurcation Topology Transfer in Electrostatically Coupled Tuning Fork Resonators

**DOI:** 10.3390/s24051563

**Published:** 2024-02-28

**Authors:** Ming Lyu, Xiang Zhi, Na Yan, Rongjian Sun

**Affiliations:** 1Institute of Transportation, Inner Mongolia University, Hohhot 010021, China; zhixiangdut@163.com (X.Z.); yannanjust@163.com (N.Y.); 2Inner Mongolia Engineering Research Center for Intelligent Transportation Equipment, Hohhot 010021, China; 3School of Mechanical Engineering, Dalian University of Technology, Dalian 116024, China; rongjian@mail.dlut.edu.cn

**Keywords:** mode localization, experimental characterization, bifurcation topology transfer

## Abstract

Bifurcation topology transfer phenomena in the presence of mode localization are investigated using double-ended fixed electrostatically coupled tuning fork resonators. An analytical model is proposed for the coupled tuning fork resonators, and the effects of feedthrough capacitance on the structure are also analyzed and eliminated by means of data post-processing. Then, an open-loop experimental platform is established, when the system is in balance state, the quality factor is obtained under test as Q = 9858, and comparison of the experiment with numerical simulation results is in good agreement. Finally, with the voltage increases, the two resonators gradually exhibit nonlinear characteristics. It is worth noting that when one of the coupled resonators exhibits nonlinear vibration behavior, even though the vibration amplitude of the other resonator is lower than the critical amplitude, it still exhibits nonlinear behavior, and the results confirm the existence of the bifurcation topology transfer phenomenon in coupled resonators’ mode localization phenomenon.

## 1. Introduction

In recent years, coupled array structures have become a development trend in the configuration design of micro mechanical resonant sensors [1,2], which can improve the sensitivity of sensors and achieve multifunctional detection [3]. However, as resonator structures tend to miniaturize and micro resonators are prone to exhibit nonlinear vibration behavior under size effects, while electrostatically coupled resonators exhibit more complex nonlinear behavior. Therefore, exploring the nonlinear dynamics of electrostatically coupled resonators has great research value for the practical applications of micro resonant sensors. The phenomenon of mode localization arises from coupled array structures (it is present in systems containing multiple resonators (at least two), usually coupled to each other via electrostatic or mechanical coupling, and the output metric is usually based on the amplitude-dependent amount of variation of the multiple resonators.). Sensors designed using this phenomenon have high sensitivity and good anti-interference ability, and have been widely used in device design, including micro mass sensors [4,5], acceleration sensors [6,7], and electrometers [8].

For mode localization sensors, the vibration amplitude can be improved by increasing the excitation voltage, thereby improving the signal-to-noise ratio. It is worth noting that as the excitation voltage increases, the coupled resonator exhibits nonlinear vibration behavior (stiffness softening or stiffness hardening). At present, there is a lack of understanding of the complex nonlinear phenomena in electrostatically coupled resonators, especially the impact mechanism of nonlinear coefficients on the vibration behavior of electrostatically coupled resonators under different excitation conditions. In [9], the nonlinear dynamic behavior of electrostatically coupled resonators containing up to fifth-order nonlinear terms under parametric excitation was investigated, and the simulation results shown that the fifth-order nonlinearity will cause a complexity in bifurcation topology and an increase in the total number of multimodal solutions. Li et al. [10] conducted theoretical research on the nonlinear dynamic behavior and corresponding parameter identification of a pair of coupled rigid beams. Through bifurcation analysis, it was found that there may be discontinuous frequency response characteristics in the coupled system, but no experiments were conducted. The coupled resonators are affected by static electricity, geometry, and other nonlinear forces at the micro scale, which may cause chaos phenomenon. Luo et al. [11] studied the chaotic behavior of coupled resonators with analog circuits, and controlled the chaotic behavior through adaptive control methods; the effectiveness of the control method was demonstrated through circuit experiments. Currently, the nonlinear research on coupled resonators mainly adopts theoretical simulations to study fixed beam structures, but in practical sensor applications, tuning fork beam resonators are often used as the sensitive structures of resonant sensors. Wang et al. [12] achieved frequency comb phenomenon using two electrostatically coupled tuning fork beam structures under 1:3 internal resonance. Guo et al. [13,14] applied the mode localization phenomenon generated by a pair of mechanically coupled tuning fork beams as a sensitive mechanism to the design of airflow sensors. In addition, the team led by Chang‘s designed a series of sensors using a coupled tuning fork beam structure to generate mode localization phenomena, including a voltmeter [15], electrometer [16], current sensor [17], etc. To summarize, the coupled tuning fork structure was applied many times in resonant sensors. However, with the miniaturization of the resonator’s size, smaller excitation will cause the resonator to exhibit nonlinear vibration behavior, and the influence of nonlinear forces cannot be ignored. Thus, exploring the nonlinear dynamic characteristics of electrostatically coupled tuning fork beam structures can provide technical support for their design and application in sensors. Utilizing the nonlinear jumping property, it is promising to apply the phenomenon to the design of threshold switches.

This paper focuses on the typical configuration of electrostatically coupled tuning fork resonators in mode localization phenomena, and conducts in-depth research on the bifurcation topology transfer phenomenon in electrostatically coupled resonators on their vibration behaviors. Firstly, the designed devices are processed based on the standard process of SOI, and the printed circuit board (PCB) of the test circuit is designed using Altium Designer software 17 (Altium Corporation, Sydney, Australia) to connect external circuits with the tested device. Secondly, the working principles of electrostatic drive and capacitance detection are elucidated, and the mechanical model in the system is transformed into an electrical model through equivalent circuit principles, thereby revealing the influence of parasitic feedthrough capacitance on the detection signal. Finally, the results is confirmed by experimental measurements that existence of bifurcation topology transfer phenomenon in the mode localization phenomenon of coupled resonators.

## 2. Structure Design

Taking typical electrostatically coupled resonant structures as the research object, we delve into their complex vibration behaviors under different excitations. As shown in Figure 1, the model consists of two electrostatically coupled resonators, which are composed of two end tuning fork beams and driven by fixed electrodes. Here, only the resonators are active, and the rest of the structures are anchored. The structural parameters are listed in Table 1.

A reduced-order model consisting of two coupled ordinary differential equations (ODEs) can be generated using the Galerkin discretization to describe the dynamics of two resonators.
(1)$$\left\{\begin{array}{l} EI\frac{\partial^{4}{\tilde{w}}_{1}\left(\tilde{x},\tilde{t}\right)}{\partial {\tilde{x}}^{4}}+\rho A\frac{\partial^{2}{\tilde{w}}_{1}\left(\tilde{x},\tilde{t}\right)}{\partial {\tilde{t}}^{2}}+\tilde{c}\frac{\partial {\tilde{w}}_{1}\left(\tilde{x},\tilde{t}\right)}{\partial \tilde{t}}-\frac{EA}{2l}{\int_{0}^{l}\left(\frac{\partial {\tilde{w}}_{1}\left(\tilde{x},\tilde{t}\right)}{\partial \tilde{x}}\right)}^{2}d\tilde{x}\frac{\partial^{2}{\tilde{w}}_{1}\left(\tilde{x},\tilde{t}\right)}{\partial {\tilde{x}}^{2}}\\ =\frac{1}{2}\frac{\varepsilon_{0}bH_{1}\left(\tilde{x}\right){\left(V_{dc1}+V_{ac1}\cos\left(\tilde{\Omega }\tilde{t}\right)\right)}^{2}}{{\left(g-{\tilde{w}}_{1}\left(\tilde{x},\tilde{t}\right)\right)}^{2}}-\frac{1}{2}\frac{\varepsilon_{0}bH_{2}\left(\tilde{x}\right)V_{s1}^{2}}{{\left(g+{\tilde{w}}_{1}\left(\tilde{x},\tilde{t}\right)\right)}^{2}}-\frac{1}{2}\frac{\varepsilon_{0}bH_{1}\left(\tilde{x}\right)V_{c}^{2}}{{\left(g+{\tilde{w}}_{1}\left(\tilde{x},\tilde{t}\right)-{\tilde{w}}_{2}\left(\tilde{x},\tilde{t}\right)\right)}^{2}}\\ EI\frac{\partial^{4}{\tilde{w}}_{2}\left(\tilde{x},\tilde{t}\right)}{\partial {\tilde{x}}^{4}}+\rho A\frac{\partial^{2}{\tilde{w}}_{2}\left(\tilde{x},\tilde{t}\right)}{\partial {\tilde{t}}^{2}}+\tilde{c}\frac{\partial {\tilde{w}}_{2}\left(\tilde{x},\tilde{t}\right)}{\partial \tilde{t}}-\frac{EA}{2l}{\int_{0}^{l}\left(\frac{\partial {\tilde{w}}_{2}\left(\tilde{x},\tilde{t}\right)}{\partial \tilde{x}}\right)}^{2}d\tilde{x}\frac{\partial^{2}{\tilde{w}}_{2}\left(\tilde{x},\tilde{t}\right)}{\partial {\tilde{x}}^{2}}\\ =-\frac{1}{2}\frac{{\varepsilon_{0}bH_{1}\left(\tilde{x}\right)V_{dc2}}^{2}}{{\left(g+{\tilde{w}}_{2}\left(\tilde{x},\tilde{t}\right)\right)}^{2}}+\frac{1}{2}\frac{\varepsilon_{0}bH_{2}\left(\tilde{x}\right)V_{s2}^{2}}{{\left(g-{\tilde{w}}_{2}\left(\tilde{x},\tilde{t}\right)\right)}^{2}}+\frac{1}{2}\frac{\varepsilon_{0}bH_{1}\left(\tilde{x}\right)V_{c}^{2}}{{\left(g+{\tilde{w}}_{1}\left(\tilde{x},\tilde{t}\right)-{\tilde{w}}_{2}\left(\tilde{x},\tilde{t}\right)\right)}^{2}} \end{array}\right.$$
(2)$$\left\{\begin{array}{l} H_{1}\left(\tilde{x}\right)=\left|H\left(\tilde{x}-\frac{l-l_{e1}}{2l}\right)-H\left(\tilde{x}-\frac{l+l_{e1}}{2l}\right)\right|,\\ H_{2}\left(\tilde{x}\right)=\left|H\left(\tilde{x}-\frac{l-l_{e1}}{2l}\right)-H\left(\tilde{x}-\frac{l-l_{e1}+2l_{e2}}{2l}\right)+H\left(\tilde{x}-\frac{l+l_{e1}-2l_{e2}}{l}\right)-H\left(\tilde{x}-\frac{l+l_{e1}}{l}\right)\right| \end{array}\right.$$

In Equation (2), the first term on the right side represents the electrostatic force driving the electrode, and the second term on the right side of the equation represents the electrostatic force generated by the coupling voltage. From left to right, the left side of the equation sequentially represents the influence of elastic force, inertial force, damping force, geometric nonlinear force, and axial stress caused by the elongation of the neutral plane of the beam. The Heaviside function is used to represent the distribution of the electrode.

For ease of calculation, the following variables are introduced to make the equation dimensionless, the electrostatic force term is subjected to third-order Taylor expansion and is finally solved using the multiscale method, and the specific solution process can be referred to in [6].
(3)$$\left\{\begin{array}{l} \frac{\partial^{4}w_{1}}{\partial x^{4}}+\frac{\partial^{2}w_{1}}{\partial t^{2}}+c\frac{\partial w_{1}}{\partial t}-\alpha_{1}{\int_{0}^{1}\left(\frac{\partial w_{1}}{\partial x}\right)}^{2}dx\frac{\partial^{2}w_{1}}{\partial x^{2}}=\frac{\alpha_{2}H_{1}{\left(V_{dc1}+V_{ac1}\cos\left(\Omega t\right)\right)}^{2}}{{\left(1-w_{1}\right)}^{2}}-\frac{\alpha_{2}H_{2}V_{s1}^{2}}{{\left(1+w_{1}\right)}^{2}}-\frac{\alpha_{2}H_{1}V_{c}^{2}}{{\left(1+w_{1}-w_{2}\right)}^{2}}\\ \frac{\partial^{4}w_{2}}{\partial x^{4}}+\frac{\partial^{2}w_{2}}{\partial t^{2}}+c\frac{\partial w_{2}}{\partial t}-\alpha_{1}{\int_{0}^{1}\left(\frac{\partial w_{2}}{\partial x}\right)}^{2}dx\frac{\partial^{2}w_{2}}{\partial x^{2}}=-\frac{\alpha_{2}H_{1}V_{dc2}^{2}}{{\left(1+w_{2}\right)}^{2}}+\frac{\alpha_{2}H_{2}V_{s2}^{2}}{{\left(1-w_{2}\right)}^{2}}+\frac{\alpha_{2}H_{1}V_{c}^{2}}{{\left(1+w_{1}-w_{2}\right)}^{2}} \end{array}\right.$$
(4)$$\begin{array}{l} {\tilde{w}}_{1}\left(0,t\right)={\tilde{w}}_{1}\left(l,t\right)=\frac{\partial {\tilde{w}}_{1}}{\partial x}\left(0,t\right)=\frac{\partial {\tilde{w}}_{1}}{\delta x}\left(l,t\right)\\ {\tilde{w}}_{2}\left(0,t\right)={\tilde{w}}_{2}\left(l,t\right)=\frac{\partial {\tilde{w}}_{2}}{\partial x}\left(0,t\right)=\frac{\partial {\tilde{w}}_{2}}{\delta x}\left(l,t\right) \end{array}$$
(5)$$w_{1}=\frac{{\tilde{w}}_{1}}{g_{a}},w_{2}=\frac{{\tilde{w}}_{2}}{g_{a}},x=\frac{\tilde{x}}{l},t=\frac{\tilde{t}}{\tau },\tau =\sqrt{\frac{\rho Al^{2}}{EI}}$$
(6)$$c=\frac{l^{2}\tilde{c}}{\sqrt{EIA\rho }},\alpha_{1}=6{\left(\frac{g}{h}\right)}^{2},\alpha_{2}=\frac{\varepsilon_{0}bl^{5}}{2EIg^{3}},\Omega =\tilde{\Omega }\tau ,$$
(7)$$\begin{array}{l} w_{1}\left(0,t\right)=w_{1}\left(1,t\right)=\frac{\partial w_{1}}{\partial x}\left(0,t\right)=\frac{\partial w_{1}}{\delta x}\left(1,t\right)\\ w_{2}\left(0,t\right)=w_{2}\left(1,t\right)=\frac{\partial w_{2}}{\partial x}\left(0,t\right)=\frac{\partial w_{2}}{\delta x}\left(1,t\right) \end{array}$$

## 3. Device Processing and Testing Circuits

The entire structure is manufactured using the standard SOI (silicon-on-insulator) process, which has achieved a relatively mature commercialization process in MEMSs’ (micro-electro-mechanical systems’) device processing. The SOI wafer parameters used are shown in Table 2. Figure 2a contains the flowchart of the SOI manufacturing process, which mainly includes five steps.

Figure 2b is the structural diagram of the coupled resonators; it can be seen that the perpendicularity of the side walls processed by the device is good, and the tuning fork beam structure used in this paper is a solid structure. This paper will conduct experimental research on the nonlinear mode localization characteristics of the proposed electrostatically coupled resonator structure by building an experimental platform. Due to the limitations in processing technology, the driving gap *g_a_* = 3 μm, coupling gap *g_c_* = 3 μm, resonator length *l* = 600 μm, resonator thickness *h* = 10 μm, the detection electrode, and driving electrode are equal to *l_e_*_1_ = 250 μm.

The dynamic current *i_in_* generated by the vibration of the resonator is usually at nA~μA, and it is difficult to detect directly; therefore, it is necessary to amplify the detected signal. Here, a transimpedance amplifier is used to amplify and convert the weak current signal’s output on the resonator’s detection electrode into a voltage signal. The schematic diagram of the transimpedance amplification circuit is shown in Figure 3b, and the amplifier used is a low-noise amplifier OPA657; the amplification factor is determined by the resistance *R_f_*. According to the working principle of this module, the circuit board design software Altium Designer was used to design the final PCB (printed circuit board) physical board, as shown in Figure 3a.

The electrostatic coupling resonator is equivalent to a capacitor between the driving and detection electrodes; thus, there is inevitably a capacitive parasitic feedthrough capacitor. Due to the parasitic effect between the driving electrodes and the detection electrodes, which generates additional capacitance, the presence of feedthrough capacitance causes the resonant frequency points of the two resonators not to coincide, which leads to errors in the measurement of amplitude ratios and seriously affects the accuracy of weakly coupled resonant transducers. Parasitic feedthrough capacitance is an inherent obstacle to all electrical interface micron-level resonant devices. The experimental detection platform of the entire coupled resonator belongs to an electromechanical integrated system, which includes both electrical system parts, such as excitation and detection, as well as mechanical system parts, such as the coupled resonator. To analyze the impacts of parasitic feedback signals on the output signal, it is necessary to convert the mechanical model in the system into an electrical model and then analyze the dynamic response of the resonator in the unified electrical system, as shown in Figure 4. The derivation of the mathematical model for feedthrough can be found in the Appendix A; the specific process can be seen in [18].

The amplitude frequency response of the coupled resonator at different feedthrough is shown in Figure 5. When the feedthrough capacitance is zero, i.e., *C_f_*_1_ = 0, *C_f_*_2_ = 0, the capacitive transducer only performs electromechanical conversion, and the peak amplitude is the maximum gain of mechanical resonance. When the feedthrough capacitor *C_f_*_1_ is not zero, i.e., *C_f_*_1_ = *C_c_*, *C_f_*_2_ = 0, it can be seen that the amplitude and frequency of the resonators undergo significant changes in the in-phase mode, especially when the amplitude decreases sharply. When the feedthrough capacitor *C_f_*_2_ is not zero, i.e., *C_f_*_1_ = 0, *C_f_*_2_ = *C_c_*, the feedthrough capacitor *C_f_*_2_ only affects resonator 2, and the amplitude of resonator 2 undergoes information distortion in both modes due to the influence of the feedthrough capacitor.

According to the analysis in Figure 5, the presence of parasitic feedthrough capacitors can cause significant distortion of resonance amplitude information. Therefore, in experimental detection, the influence of parasitic feedthrough capacitance needs to be removed. The common methods for removing feedthrough include online capacitance removal and data post-processing removal. The method of removing parasitic feedthrough through online capacitors is to use adjustable capacitors to match the parasitic feedthrough capacitance of the device, and then eliminate the feedthrough signal through differential elimination.

Another method is to remove the influence of parasitic feedthrough capacitance through data post-processing. Here, this method was used to remove the parasitic feedthrough capacitance. First, the frequency response curve of the coupled resonator is tested through the open-loop test, the results obtained include both the response signal of the electrostatically coupled resonator and the parasitic signal, as shown by the blue line in Figure 6a,c. At this time, the output test signal includes both the real signal and the parasitic signal. Next, remove the bias voltage and only retain the AC signal for the same frequency sweep process. Since the resonator is not working at this time, the obtained results only include the parasitic feedthrough signal, as shown in the red lines in Figure 6a,c. Finally, by calculating the amplitude data obtained from two frequency sweeps, the parasitic feedthrough signal is removed (the program for removing the parasitic feedthrough signal can be seen in the attachment), so that the obtained response only includes the motion signal, as shown in the graphs in Figure 6b,d, which are the true amplitude frequency and phase frequency curves after removing the parasitic signal.

## 4. Discussion

As shown in Figure 7, the built experimental equipment mainly includes a vacuum chamber, transimpedance amplifier (OPA657 (operational amplifier), Texas Instruments, Dallas, TX, USA), DC power supply, and Zurich Instruments HF2LI (Zurich, Switzerland). Moreover, the structure and the fixed PCB board are placed together in the vacuum chamber and are connected with the external DC power supply and the transimpedance amplifier through the BNC (bayonet nut connector, Specialty Digital Company, Shenzhen, China) adapter on the vacuum chamber wall using the SMA (SubMiniature version A, Specialty Digital Company, Shenzhen, China) connection line; the signal output port of the lock-in amplifier outputs the AC drive signal, which acts on the drive electrode through the coupling capacitor and the bias voltage. The dynamic current generated by the resonator’s vibration changes from the detection electrode to the voltage signal through the transimpedance amplifier, and then is input to the signal input port of the lock-in amplifier to obtain the amplitude frequency response of the electrostatically coupled resonator in the open-loop test.

The amplitude frequency response of the electrostatically coupled resonator under different coupling voltages can be obtained by building a vacuum open-loop test circuit. The open-loop test scheme of the electrostatically coupled resonator is shown in Figure 8.

After selecting bias voltage *V_dc_*_1_ = 15 V, *V_dc_*_2_ = 15 V, coupling voltage *V_c_* = 15 V, and AC driving voltage *V_ac_* = 10 mV, the frequency of the electrostatic coupling resonator is swept using the established open-loop testing scheme. As shown in Figure 9a,b, the output signals of resonators 1 and 2 with feedthrough in the open-loop test are shown. Through the outputs of the two resonators, it can be observed that, under electrostatic coupling, mode 1 is out of phase and mode 2 is in phase. Due to the resonator’s capability of being treated as a capacitor, the presence of parasitic feedthrough capacitors is inevitable. The presence of parasitic feedthrough capacitors can cause a significant distortion of resonance amplitude information. The experimental data after removing the feedthrough capacitor using data post-processing are shown in Figure 9c,d, and the quality factor is obtained during testing as Q = 9858. By comparing the experimental values with theoretical simulation calculation data, it can be found that the two are in good agreement.

As shown in Figure 10, the solid line represents a forward frequency sweep and the dotted line represents a reverse frequency sweep; 10-10-35-R represent *V_dc_*_1_-*V_dc_*_2_-*V_c_*, respectively, and R represents the reverse frequency sweep. When maintaining the AC driving voltage *V_ac_*_1_ = 100 mV and coupling voltage difference equal to 25 V unchanged, adjust the magnitude of the bias voltage *V_dc_*_1_ and *V_dc_*_2_. As the bias voltage continues to increase, the amplitudes of the two resonators also continue to increase and gradually exhibit nonlinear characteristics. It is worth noting that when one resonator exhibits nonlinear vibration behavior in a coupled resonator, there is a bifurcation topology phenomenon between the two resonators. This phenomenon refers to when one of the coupled resonators exhibits nonlinear vibration, and even if the vibration amplitude of the other resonator is lower than the critical amplitude in the equilibrium state, it will still exhibit nonlinear behavior [19,20]. The critical amplitude is the oscillation amplitude above which bistability occurs. Thus, it is the transition amplitude from the linear to the nonlinear behavior [21]. Such as when the bias voltage is equal to 10-10-35, the amplitude of *w*_2_ is equal to 0.18 mV, and its amplitude is lower than the critical amplitude and exhibits linear vibration, as shown in Figure 10b. With the bias voltage increased to 20-20-45, resonator 2 exhibits nonlinear vibration, and the amplitude of resonator 1 is 0.15 mV, which is significantly lower than the critical amplitude. However, it still exhibits nonlinear vibration behavior, as shown in Figure 10a. Therefore, this proves that there is a significant topological bifurcation transfer phenomenon in the mode localization phenomenon of coupled resonators.

## 5. Conclusions

In this paper, a pair of electrostatically coupled tuning fork beams were studied, and their nonlinear characteristics were explored by constructing an experimental platform. In addition, it was found that a phenomenon of bifurcation topology transfer occurs. The main conclusions are listed as follows: (i) A theoretical model for electrostatically coupled tuning fork beam structure was established. (ii) The designed devices were processed based on the standard process of SOI, and the PCB of the test circuit was designed to connect tested devices. (iii) The working principles of electrostatic drive and capacitance detection were elucidated, and the mechanical model in the system was transformed into an electrical model through equivalent circuit principles, thereby revealing the influence of parasitic feedthrough capacitance on the detection signal. Finally, an open-loop testing circuit was built in a vacuum environment to test the modal localization characteristics of the electrostatic coupling resonator and the performance of the mode localization’s acceleration sensor.

## Figures and Tables

**Figure 1 sensors-24-01563-f001:**
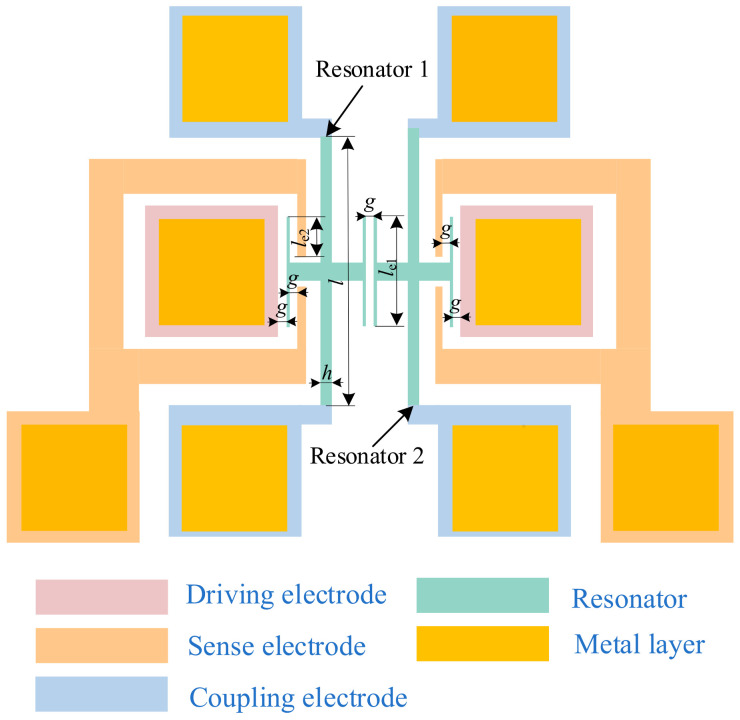
The electrostatically coupled resonator structure using mode localization phenomenon.

**Figure 2 sensors-24-01563-f002:**
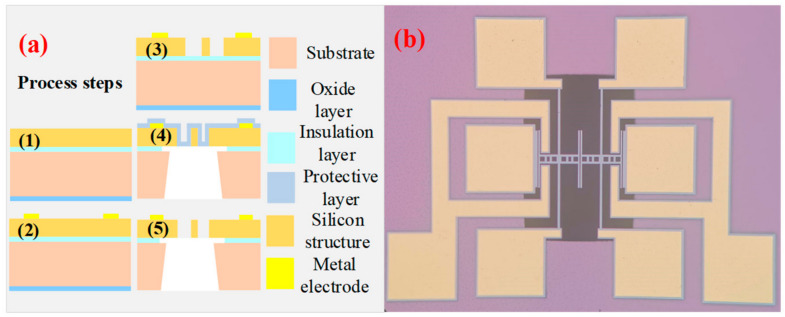
(**a**) SOI manufacturing process; (**b**) optical micro-scope image.

**Figure 3 sensors-24-01563-f003:**
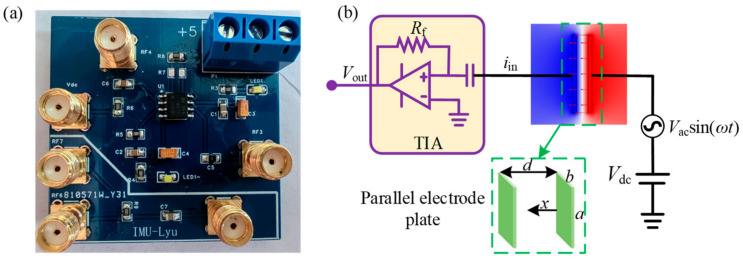
Transimpedance amplifier: (**a**) PCB board of transimpedance amplifier; (**b**) schematic diagram of amplification circuit.

**Figure 4 sensors-24-01563-f004:**
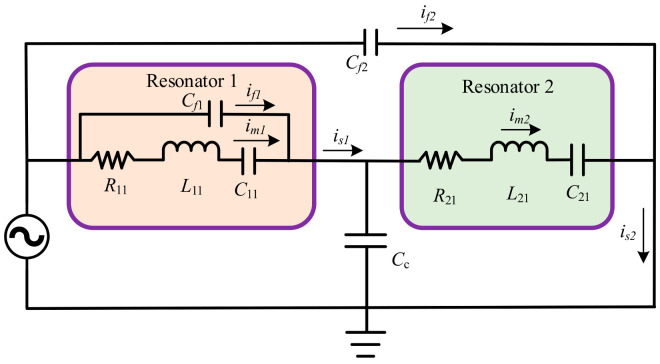
Parasitic feedthrough equivalent circuit model.

**Figure 5 sensors-24-01563-f005:**
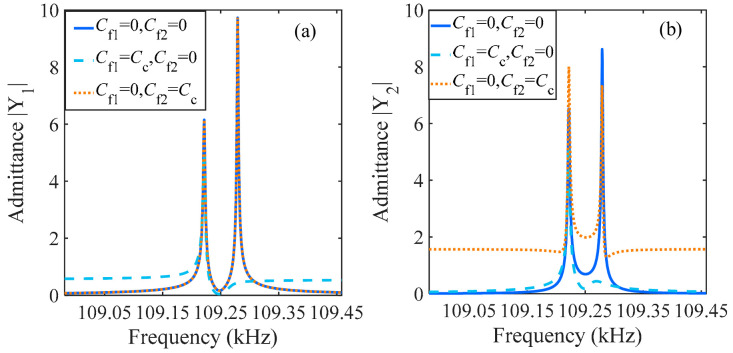
Amplitude–frequency response of the coupled resonators electrical model under different feedthrough effects: (**a**) admittance |Y_1_|; (**b**) admittance |Y_2_|.

**Figure 6 sensors-24-01563-f006:**
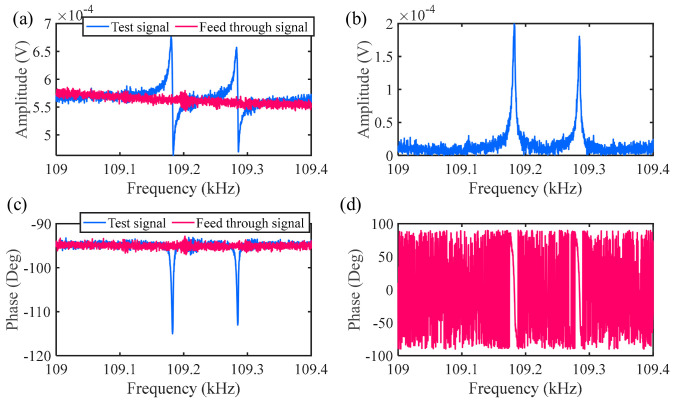
The removal of feedthrough signals: parts (**a**,**c**) are the measured amplitude−frequency and phase−frequency signals containing feedthrough signals; parts (**b**,**d**) are the real amplitude−frequency and phase−frequency motion signals that remove feedthrough signals.

**Figure 7 sensors-24-01563-f007:**
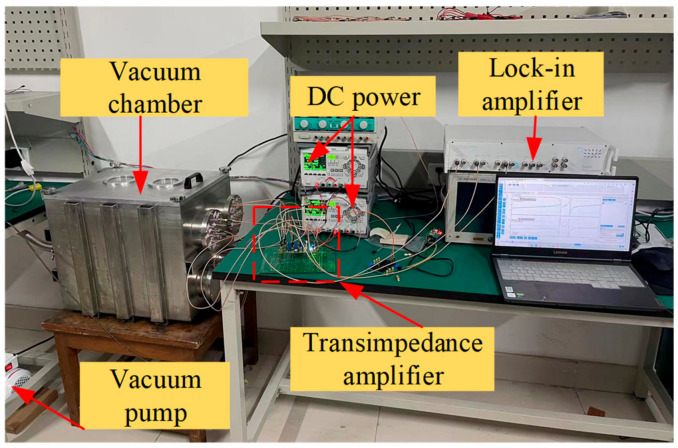
The electrostatically coupled resonator vacuum open-loop frequency response platform.

**Figure 8 sensors-24-01563-f008:**
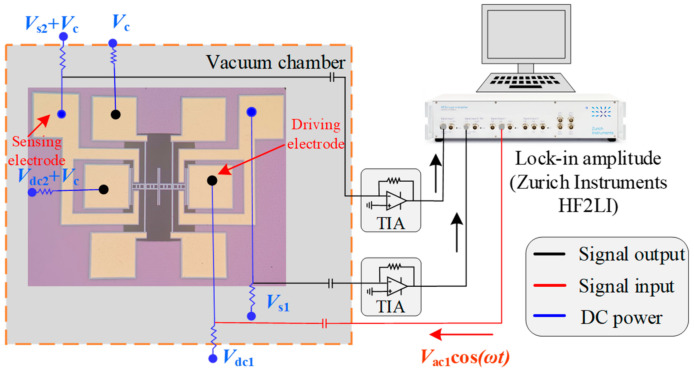
Open-loop test scheme of electrostatically coupled resonator.

**Figure 9 sensors-24-01563-f009:**
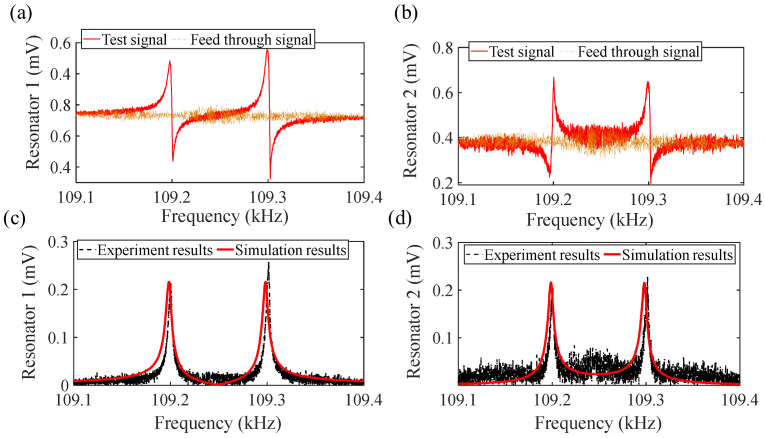
The response of the electrostatically coupled resonators under open-loop test: parts (**a**,**b**) are the signals of resonators 1 and 2 with feedthrough; parts (**c**,**d**) are the amplitudes of resonators 1 and 2 after feedthrough.

**Figure 10 sensors-24-01563-f010:**
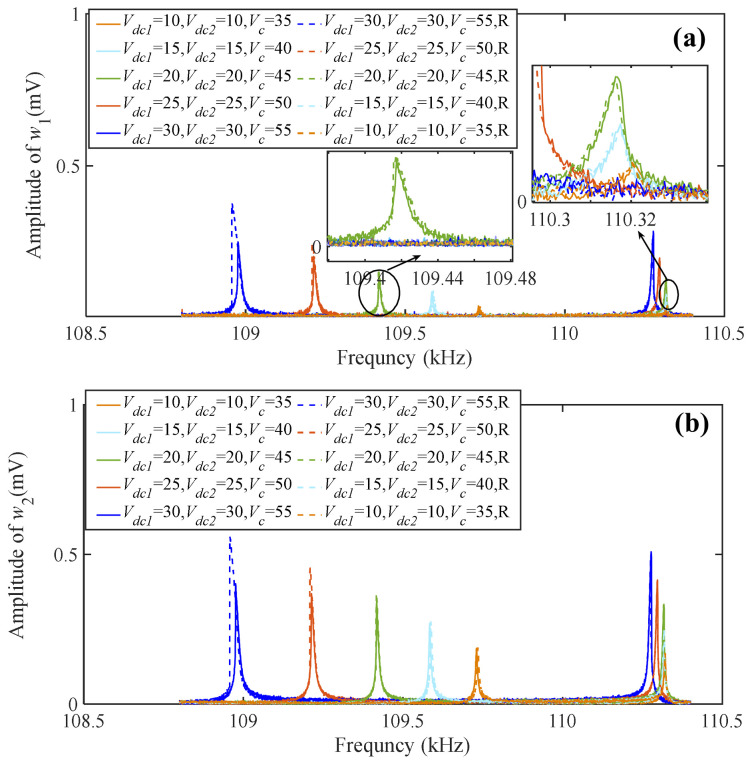
Variation of resonator amplitude–frequency characteristics under different bias voltages: (**a**) resonator 1; (**b**) resonator 2.

**Table 1 sensors-24-01563-t001:** Structure parameters.

Type	Parameter	Value
Length of microbeam	*l*	600 μm
Young’s modulus	*E*	169 GPa
Microbeam height	*h*	10 μm
Microbeam width	*b*	25 μm
Density	*ρ*	2320 kg/m^3^
Air gap	*g*	3 μm
Air gap	*g*	3 μm
Dielectric constant	*ε_0_*	8.85 × 10^−12^ F/m
Silicon structure thickness	25 ± 1 μm	
Surface metal thickness	≤5E0	
Thickness of buried oxygen layer	1 μm ± 5%	
Substrate thickness	550 ± 20 μm	

**Table 2 sensors-24-01563-t002:** SOI wafer parameters (4 inches).

Name	Specifications
Crystal growth method	CZ
Crystallographic orientation	100
Dopant type	P/B
Resistivity	0.01–0.02 ohm.cm
Main unit location	110
Silicon structure thickness	25 ± 1 μm
Surface metal thickness	≤5E0
Thickness of buried oxygen layer	1 μm ± 5%
Substrate thickness	550 ± 20 μm

## Data Availability

Data are contained within this article.

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
