# Peer review of "Experimental Investigation of Mode Localization’s Bifurcation Topology Transfer in Electrostatically Coupled Tuning Fork Resonators"

_sensors, 2024, doi:10.3390/s24051563_

Round 1

Reviewer 1 Report

Comments and Suggestions for Authors

This paper focuses on the typical configuration of double ended fixed electrostatic coupled 10 tuning fork resonators in mode localization phenomena. The manuscript is well constructed. The result is interesting and valuable. However, the authors are still suggested to consider the comments below:

1)- Parameters in Table 1 should be changed to italics

2)- The equation (2) should be right-aligned

3)- Please define each abbreviation used.

4)- There is a lack of clarity regarding the meaning of each parameter in Figure 10.

Comments on the Quality of English Language

Language is okay

Reviewer 2 Report

Comments and Suggestions for Authors

The primary emphasis of this study is on the conducts in-depth research on the bifurcation topology transfer phenomenon in electrostatic coupled resonators on their nonlinear vibration behavior. Bifurcation topology transformation is commonly present in mode localization sensors, this paper provides guidance to mode localized sensors. However, some comments have to be addressed prior to publication:

1) It was pointed out that the phenomenon of mode localization occurs in coupled array structures. The author can introduce it when the article first proposes the phenomenon of mode localization.

2) Kindly include information on the MEMS device, specifying its length, width, thickness and the electrical parameters.

3) Strictly speaking, the tuning fork structure should have both in-phase and antiphase modes on one resonator. In this article is obviously two coupled resonators, please clarify.

4) In Fig.1, identifying the structural layers and anchoring locations of activities will help improve readability.

5) In Fig.6, data filtering is used to remove feedthrough. However, data filtering is not suitable for real-time measurement mode. The author can illustrate the impact of feedthrough on real-time sensing process and propose solutions to remove feedthrough.

6) Legends in Fig.10 is not clear.

7) The author mentioned “the amplitude of w2 is equal to 0.18mV, its amplitude is lower than the critical amplitude and exhibits linear vibration”. However, the critical amplitude in the text is not clearly defined, which can be clearly described.

8) It is also encouraged to indicate the application of bifurcation topology transformation in sensing.

Comments on the Quality of English Language

The paper is well rewritten. 

Reviewer 3 Report

Comments and Suggestions for Authors

1. As a technical paper with experiments, numerical comparisons should appear in the abstract.

2. To increase readability, more structural parameters need to be added in Figure 1.

3. Revise the deficiency in Figure 3b.

4. More detailed derivation and description to explain impact of the feedthrough.

5. What is the quality factor of the resonator under test? The abstract mentions nonlinearity, but no significant nonlinearity is observed from the pictures of the test.

6. What is the novelty of this manuscript?

7. Is the removal of feedthroughs proposed in the manuscript feasible? The feedthrough is temperature dependent.

8. The color distinction between the curves in Figure 10 is not obvious. Too many curves.

Comments on the Quality of English Language

Specialized vocabulary and grammar need to be revised.

Round 2

Reviewer 3 Report

Comments and Suggestions for Authors

1. More  discussion and numerical comparisons for experiments in abstract! Must !

2. Where is Microbeam height, Microbeam width, Air gap in Figure 1?

3. Analysis of the feedthrough in the pickoff end are missing!Must !

4.  Novelty need mentioned in abstract!

5. The value of quality factor is 9858, which is not a high value for a MEMS resonator. The nonlinearity claimed in the manuscript is suspicious.

Comments on the Quality of English Language

No comments
